# Patterns of asymmetry and energy cost generated from predictive simulations of hemiparetic gait

**Russell T. Johnson** [1]*, **Nicholas A. Bianco** [2], **James M. Finley** [1,3,4]

**1** Division of Biokinesiology and Physical Therapy, University of Southern California, Los Angeles, California, United States of America, **2** Department of Mechanical Engineering, Stanford University, Palo Alto, California, United States of America, **3** Department of Biomedical Engineering, University of Southern California, Los Angeles, California, United States of America, **4** Neuroscience Graduate Program, University of Southern California, Los Angeles, California, United States of America

* rtjohnso@usc.edu

**Data Availability Statement:** All models and code are available at https://simtk.org/projects/post-stroke-sym.

**Funding:** JMF received support from the National Institutes of Health (https://www.nichd.nih.gov/

## Abstract

Hemiparesis, defined as unilateral muscle weakness, often occurs in people post-stroke or people with cerebral palsy, however it is difficult to understand how this hemiparesis affects movement patterns as it often presents alongside a variety of other neuromuscular impairments. Predictive musculoskeletal modeling presents an opportunity to investigate how impairments affect gait performance assuming a particular cost function. Here, we use predictive simulation to quantify the spatiotemporal asymmetries and changes to metabolic cost that emerge when muscle strength is unilaterally reduced and how reducing spatiotemporal symmetry affects metabolic cost. We modified a 2-D musculoskeletal model by uniformly reducing the peak isometric muscle force unilaterally. We then solved optimal control simulations of walking across a range of speeds by minimizing the sum of the cubed muscle excitations. Lastly, we ran additional optimizations to test if reducing spatiotemporal asymmetry would result in an increase in metabolic cost. Our results showed that the magnitude and direction of effort-optimal spatiotemporal asymmetries depends on both the gait speed and level of weakness. Also, the optimal speed was 1.25 m/s for the symmetrical and 20% weakness models but slower (1.00 m/s) for the 40% and 60% weakness models, suggesting that hemiparesis can account for a portion of the slower gait speed seen in people with hemiparesis. Modifying the cost function to minimize spatiotemporal asymmetry resulted in small increases (~4%) in metabolic cost. Overall, our results indicate that spatiotemporal asymmetry may be optimal for people with hemiparesis. Additionally, the effect of speed and the level of weakness on spatiotemporal asymmetry may help explain the well-known heterogenous distribution of spatiotemporal asymmetries observed in the clinic. Future work could extend our results by testing the effects of other neuromuscular impairments on optimal gait strategies, and therefore build a more comprehensive understanding of the gait patterns observed in clinical populations.

about/org/ncmrr; NIH NCMRR R01HD091184).
The funders had no role in the study design, data
collection and analysis, decision to publish, or
preparation of the manuscript.

**Competing interests:** The authors have declared
that no competing interests exist.

## Author summary

Hemiparesis is muscle weakness that occurs primarily on one side of the body and can
occur in a range of different clinical populations. Individuals with hemiparesis tend to
walk with an asymmetrical motion and expend more energy while walking than other
age-matched individuals. We still do not understand how each specific change to the neu-
romuscular system is linked with changes in walking patterns, in part because it is difficult
to test one change at a time in people. Instead, we can use a mathematical model of the
musculoskeletal system that represents the individual changes to the muscular system that
occur in people with hemiparesis. In this study, we modeled how a weakness on one side
of the body can impact walking patterns. We found that the level of weakness and the
walking speed affect the asymmetrical walking patterns of our models, but do not change
the total energy cost. Overall, our study is one step towards better understanding how
neuromuscular impairments affect walking patterns. This knowledge could be applied to
identify rehabilitation strategies that are most likely to improve walking in people with
hemiparesis.

## 1. Introduction

Hemiparesis is defined as a decrease in muscular strength that primarily affects one side of the
body, and it commonly occurs in clinical populations such as people post-stroke [1,2], individ-
uals with cerebral palsy [3], or after a traumatic brain injury [4,5]. In each of these cases, hemi-
paresis occurs alongside a number of other distinct neuromuscular impairments [6–8], such as
muscle spasticity [9,10], apraxia [11], or generation of abnormal patterns of muscle coordina-
tion [12–14]. The combination of these unilateral neuromuscular impairments result in
marked gait deviations: for example, people post-stroke walk slower, spend more time in dou-
ble support, and typically display asymmetrical step lengths, step times, and joint kinematics
[15–17]. In addition to slower preferred gait speeds and spatiotemporal asymmetries, people
post-stroke also walk with a greater metabolic cost of transport (COT) than age- and speed-
matched control subjects [15,18]. Individuals with cerebral palsy display many of the same
broad characteristics of gait performance: walking with a slower gait speed and greater meta-
bolic COT [19–21] as well as spatiotemporal asymmetries [22–24]. Determining causal links
between neuromuscular impairments and gait performance is challenging because it is impos-
sible to independently modulate all known impairments in people with hemiparesis and then
observe the effects of these changes on gait mechanics or metabolic cost. Yet, an ongoing
objective for clinicians and researchers is to determine the causal associations relating neuro-
muscular impairments, gait deviations, and the energetic cost of walking [15,25–28]. Better
knowledge of how these characteristics interact could be used to identify better intervention
targets that are most likely to improve rehabilitation outcomes.

 Previous researchers have tried to identify how specific neuromuscular impairments relate
to measures of gait performance by using regression or correlation analyses. Hemiparesis in
the knee extensors and hip flexors after a stroke has been correlated with reductions in gait
speed and increases in step length asymmetries [25–28]. Similarly, spatiotemporal asymme-
tries have been shown to correlate positively with the energy cost of walking post-stroke [15].
Despite the abundance of work examining relationships between impairment and measures of
gait performance post-stroke, one key limitation of these approaches is that regression analyses
are insufficient to determine the causal effects of any specific impairment.

When neurotypical individuals walk with greater spatiotemporal asymmetry, their metabolic COT increases compared to symmetrical walking patterns [29,30]. With this rationale, common rehabilitation interventions have targeted these spatiotemporal asymmetries to try to improve measures of gait performance in people with hemiparesis. However, recent studies using biofeedback on step length asymmetry have shown little or no improvement in metabolic COT when people post-stroke walk with more symmetric step lengths [31–35]. Additionally, improving muscle strength is a common target for rehabilitation protocols as a way to improve spatiotemporal asymmetries, however it's still unclear if improvements in muscle strength lead to improvements in gait performance [36,37].

Musculoskeletal modeling and predictive simulation provide the opportunity to generate predictions about how specific neuromuscular impairments affect gait performance as researchers can modify muscle parameters in a systematic way while keeping other parameters unchanged [38,39]. Previous researchers have used musculoskeletal modeling and predictive simulation to predict gait patterns for both healthy and clinical populations, such as people with cerebral palsy or people who walk with lower limb prosthetics [38,40–42]. For example, these studies have helped identify that weak plantar flexor muscle groups can explain calcaneal or "heel-walking" gait patterns, as seen in children with spastic diplegia [38,43]. Overall, predictive simulations have led to valuable insight into principles of motor control [40,44,45], the effect of impairments on gait mechanics and energetics [38,39,43,46] and the effect of gait patterns on joint loading [47,48]. These studies, along with recent advancements in both computational efficiency and accessibility [49–51], have allowed predictive modeling and simulation studies to help reveal important principles of gait mechanics [52]. Therefore, we can apply the methodology of musculoskeletal modeling and predictive simulation to understand the optimal gait patterns for people with neuromuscular impairments.

The aim of this study was to use predictive simulations to explore how a common neuromuscular impairment, hemiparesis, should impact patterns of spatiotemporal asymmetry and metabolic cost for gait patterns that minimize muscular effort. To simulate hemiparesis in our musculoskeletal model, we systematically reduced the peak isometric muscle strength for all the muscles on the left limb of our model while keeping the right limb constant and evaluated patterns of spatiotemporal asymmetry and metabolic cost that emerged across a wide range of speeds while minimizing muscle excitations. We hypothesized that asymmetric walking patterns would emerge as being optimal as we increased the magnitude of hemiparesis. We then asked how enforcing symmetric step lengths and step times in models with simulated hemiparesis impacted the metabolic cost of transport. Here, we expected that enforcing symmetry would lead to marked increases in the metabolic cost of transport when compared to models where symmetry was not enforced. Overall, this work will allow us to gain insight into how a specific neuromuscular impairment impact gait deviations and the metabolic energy cost of walking, independent of other neuromuscular changes that occur in a range of clinical populations.

## 2. Methods

### 2.1. Musculoskeletal model

A two-dimensional, sagittal-plane musculoskeletal model with 11 degrees-of-freedom (DOF) was used (Fig 1A) within the OpenSim Moco software to generate optimal control simulations of walking [49]. The model had a pelvis that translated and rotated relative to the ground with 3 DOF and the torso was rigidly attached to the pelvis. Each hip, knee, ankle, and toe joints were each modeled as 1 DOF pin joints. The model was actuated with 24 Hill-type muscle-tendon units (12 per limb) based on the *DeGrooteFregly2016Muscle* model with compliant series

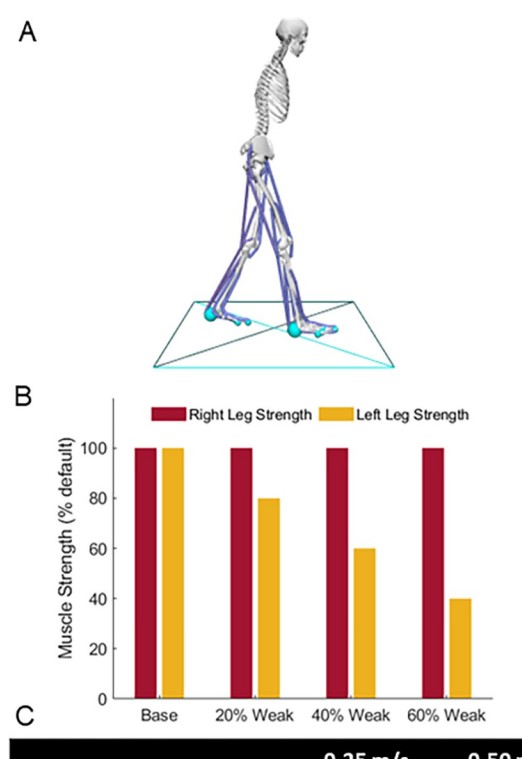

A) Two-dimensional musculoskeletal model with 11 mechanical degrees-of-freedom and 24 muscle-tendon units [86]. Blue spheres represent foot-contact elements on the model with the blue box visually representing the floor.

B) Maximum isometric muscle forces relative to base model for the 20%, 40% and 60% weak models.

D) Minimize Cost:

$$J = \frac{\sum_{i=1}^{24} \int_0^{t_f} e_i^3(t)\, dt}{q_{pelx}(t_f) - q_{pelx}(0)}$$

Subject to:
- Model dynamics and constraints
- boundary constraints
- average speed

With respect to:
- bounds on states and controls

Solver Settings:
- 101 mesh intervals
- Hermite-Simpson Transcription
- Select Initial Guess

|  | 0.25 m/s | 0.50 m/s | 0.75 m/s | 1.00 m/s | 1.25 m/s | 1.50 m/s |
|---|---|---|---|---|---|---|
| Nominal Conditions | All | All | All | All | All | Base Only |
| SL Sym. | --- | --- | 20, 40, 60% | --- | --- | --- |
| SL and ST Sym. | --- | --- | 20, 40, 60% | --- | --- | --- |

**Fig 1. Summary of set up.** A) Two-dimensional musculoskeletal model with 11 mechanical degrees-of-freedom and 24 muscle-tendon units [86]. Blue spheres represent foot-contact elements on the model with the blue box visually representing the floor. B) Maximum isometric muscle forces relative to base model for the 20%, 40% and 60% weak models, muscle strength was uniformly reduced on the left limb, C) Table showing the optimal control conditions for each speed: we used all four models for each of the speeds between 0.25 m/s and 1.25 m/s, and ran our second analysis at 0.75 m/s for the step length symmetry (SL Sym.) and step length and step time symmetry (SL and ST Sym.). D) Overview of optimization process in Moco, where the objective function (Eqs 1, 5, or 6 respectively from methods) and constraints (Eqs 2 and 3 from methods) for the problem are sent to the solver, with specified settings. Note that the objective function value seen here is for the nominal optimizations, see the methods for further details. The OpenSim image depicted is based on models available at simtk.org and the specific model and data shown here can be downloaded from https://simtk.org/projects/post-stroke-sym.

elastic elements [53]. The DeGrooteFregly2016Muscle model was used for this study since it is smooth and continuous everywhere; this is required for Moco which uses a gradient-based optimization approach to solve optimal control problems. In addition, it provides an implicit form of muscle-tendon dynamics, which is more computationally efficient and converges more reliably compared to the explicit form of muscle-tendon dynamics [53]. The foot-ground interactions were simulated using five contact spheres per foot, represented with smooth and continuous functions [54]. We used this musculoskeletal model with symmetrical strength for the base conditions, and then created three other models where the peak isometric strength of each muscle on the left limb was reduced from the base model by 20%, 40%, and 60% for the three hemiparetic conditions (Fig 1B and Table 1). This unilateral reduction in peak strength was intended to simulate the loss of force production capacity that commonly results from

**Table 1. Peak isometric strength (in N) for each of the 12 muscles for each model.** The right limb strength remained unchanged (Base values) while the left limb strength was modified for the 20%, 40%, and 60% weakness models.

| | Base/Symmetrical | 20% | 40% | 60% |
|---|---|---|---|---|
| Biceps long head | 4105 | 3284 | 2463 | 1642 |
| Biceps short head | 557 | 445.6 | 334.2 | 222.8 |
| Gluteus maximus | 4450 | 3560 | 2670 | 1780 |
| Psoas | 2448 | 1958.4 | 1468.8 | 979.2 |
| Rectus femoris | 2192 | 1753.6 | 1315.2 | 876.8 |
| Vastus Intermedius | 9593 | 7674.4 | 5755.8 | 3837.2 |
| Gastrocnemius | 4691 | 3752.8 | 2814.6 | 1876.4 |
| Soleus | 6194 | 4955.2 | 3716.4 | 2477.6 |
| Tibialis anterior | 1227 | 981.6 | 736.2 | 490.8 |
| Extensor hallucis longus | 889 | 711.2 | 533.4 | 355.6 |
| Flexor digitorum longus | 1331 | 1064.8 | 798.6 | 532.4 |
| Flexor digitorum brevis | 938 | 750.4 | 562.8 | 375.2 |

stroke. Although the weakness resulting from acute stroke results from a loss of descending drive from the brain, we would need to simulate large reductions in feasible excitations for this loss to impact the patterns of muscle activation observed in a sub-maximal task like walking. In addition, because the magnitude of a given loss of descending drive to a set of muscles is less practical to measure empirically than the resulting weakness, this approach can be more easily extended to personalize predictive simulations. Finally, we chose to reduce all muscles on the limb by the same magnitude within each condition for two reasons: First, it allows us to keep our analysis relatively simple with mild, moderate, and more severe levels of weakness across the three hemiparetic models. Second, the alternative approach, changing the strength of each joint independently, is not well-posed as it is not trivial to assess how to weaken muscles that acts across multiple joints. We chose the level of muscle weakness (i.e., 20 to 60% unilateral weakness) based on muscle torque data from Sanchez et al., 2017 [14], that showed that ambulatory post-stroke patients can have inter-limb muscle strength asymmetries of as much as 60%.

## 2.2. Optimal control problem for nominal conditions

We ran several sets of optimal control problems in Moco (full list of conditions in Fig 1C), which were solved by minimizing the sum of the integrated muscle excitations cubed divided by the horizontal displacement of the center of mass (Eq 1), which is hypothesized to represent minimizing muscle fatigue [40,55]. While it is unclear what objective function best explains features of human gait, several studies have suggested that humans select gait patterns that reduce the muscular effort, muscle fatigue, or metabolic energy associated with the task [56–61]. These studies provide evidence that effort optimization may explain why we choose certain spatiotemporal features of our gait during steady-state walking and when exposed to novel task demands.

Direct collocation was used to solve for the set of states and controls needed to produce a full gait cycle, subject to the objective function and constraints. The set of states, $x$, were 11 generalized coordinates $q(t)$ corresponding to each joint and the three degrees of freedom for the pelvis, 11 generalized velocities $u(t)$, 24 muscle activations $a(t)$, and 24 normalized tendon forces $\tilde{F}^T(t)$. The set of controls consisted of 24 muscle excitations $e(t)$ and 24 auxiliary variables representing the derivative of normalized tendon force $\dot{\tilde{F}}^T(t)$, which were necessary for

enforcing the muscle-tendon equilibrium equations in implicit form (Eq 2; 41). The states and controls were discretized on a grid of 201 evenly distributed nodes over a complete gait cycle. Optimal control problems were solved for each of the four models for speeds of 0.25, 0.50, 0.75, 1.00, and 1.25 m/s. Each problem was formulated to generate a full stride of walking by finding the set of model states, controls, and final time $t_f$, subject to the objective function (Eq 1),

$$J_1 = \frac{\sum_{i=1}^{24} \int_0^{t_f} e_i^3(t)dt}{q_{pelvis_x}(t_f) - q_{pelvis_x}(0)} \tag{1}$$

where $e_i$ is the excitation of the $i$th muscle and $q_{pelvis_x}$ is the horizontal position of the pelvis such that the denominator represents the displacement of the model during the gait cycle. We chose to minimize muscle excitations cubed based on the hypothesis that effort is an important factor for producing voluntary movements in humans, including individuals with neuromuscular impairments.

Each optimal control problem was solved by minimizing $J_1$ subject to constraints enforcing skeletal kinematics, skeletal dynamics, muscle activation dynamics, and implicit tendon compliance dynamics (Eq 2; 37,41),

$$u(t) = \dot{q}(t)$$

$$\dot{u}(t) = f(q(t), u(t), a(t), \tilde{F}^T(t)) \tag{2}$$

$$\dot{a}(t) = f(e(t), a(t))$$

$$f(a(t), \tilde{F}^T(t), \dot{\tilde{F}}^T(t)) = 0$$

Bounds were placed on the states, controls, and auxiliary variables (Eq 3).

$$q_{LB}(t) \le q(t) \le q_{UB}(t)$$

$$u_{LB}(t) \le u(t) \le u_{UB}(t)$$

$$0.001 \le e(t) \le 1 \tag{3}$$

$$0.001 \le a(t) \le 1$$

$$0 \le \tilde{F}^T(t) \le 1.8$$

$$-1000 \le \dot{\tilde{F}}^T(t) \le 1000$$

where $q_{LB}$, $q_{UB}$, $u_{LB}$ and $u_{UB}$ represent the lower and upper bounds on each kinematic value and speed, respectively. Excitations ($e(t)$) and activations ($a(t)$) were bounded between 0.001 and 1. Normalized tendon forces ($\tilde{F}^T(t)$) were bounded between 0 and 1.8 and their derivative ($\dot{\tilde{F}}^T(t)$) were bounded between -1000 and 1000. Lastly, the optimization was also subject to periodicity constraints, where the final states and controls must equal the initial states and controls, except for the horizontal position of the pelvis $q_{pelvis_x}(t)$, which must account for

horizontal translation relative to the target velocity $v$ and $t_f$ (Eq 4).

$$q(t_f) = q(0)$$

$$q_{pelvis_x}(t_f) = q_{pelvis_x}(0) + (v * t_f)$$

$$u(t_f) = u(0) \qquad (4)$$

$$e(t_f) = e(0)$$

$$a(t_f) = a(0)$$

$$\tilde{F}^T(t_f) = \tilde{F}^T(0)$$

The final time of the gait cycle was allowed to vary within the optimization between 0.3 and 2 seconds, which allowed the optimization to use the optimal stride length and stride time which satisfied the target gait speed for each condition (Fig 1D).

We used three different initial guesses for each of the 20 conditions (four models and five speeds each). The first initial guess for each condition was the result of a tracking optimization which used averaged gait data from healthy control subjects walking at 1.4 m/s [44]. The reference data were normalized to 101 time points to align the data with the collocation points of the optimizations. The first initial guesses for each speed were generated by minimizing a weighted combination of the total integrated muscle excitations cubed and the difference between experimental and simulated kinematics and ground reaction forces (Eq 5), while matching the average gait speed (i.e., 0.25, 0.50, 0.75, 1.00, and 1.25).

$$J_{track} = J_1 + \int_0^{t_f} \left[ \left( w_1 * \sum_{i=1}^{11} \left( \frac{q_i(t) - \hat{q}_i(t)}{\sigma_{qi}} \right)^2 + \left( \frac{u_i(t) - \hat{u}_i(t)}{\sigma_{ui}} \right)^2 \right) + \left( w_2 * \sum_{j=1}^{4} \left( \frac{GRF_j(t) - G\hat{R}F_j(t)}{\sigma_j} \right)^2 \right) \right] dt \quad (5)$$

For Eq 5, the term $J_1$ represents the muscular effort (Eq 1). The second and third terms represent deviations from the experimental kinematics and ground reaction forces (GRFs), respectively. $q_i(t)$ and $u_i(t)$ are the values and velocities of model coordinate $i$ at time $t$, $\hat{q}_i(t)$ and $\hat{u}_i(t)$ are the reference kinematic data from Miller (2014) [44] of coordinate $i$ at time $t$, and $\sigma_{qi}$ and $\sigma_{ui}$ are the standard deviations of the reference data, averaged across the stride. $GRF_j(t)$ represents the model ground reaction forces at time $t$ for the horizontal and vertical forces for the left and right side, and $G\hat{R}F_j(t)$ and $\sigma_j$ represent the reference ground reaction force data means and standard deviations for each direction and limb. This initial guess was used for each of the four models for the corresponding walking speed (e.g., the tracking solution for 0.75 m/s was used as an initial guess for each of the four models at 0.75 m/s speed). While the reference data from Miller (2014) [44] was at a different gait speed than the speeds used in this study, the tracking problem solutions produced realistic gait mechanics across speeds and were reasonable initial guesses for our predictive simulations.

The other two initial guesses were chosen heuristically from solutions of other conditions, either different speeds or from different simulated hemiparesis models. For example, the 20% weak model at 0.75 m/s had the following initial guesses: (1) tracking solution at a speed of 0.75, (2) the solution to the 20% weak model at 1.00 m/s, and (3) the 40% weak solution at 0.75 m/s. If the solutions differed between initial guesses, we chose the optimal solution for each

condition with the least objective function value, and this solution was then used to calculate the spatiotemporal asymmetry and metabolic cost (see Section 2.4).

## 2.3. Step time and step length symmetry conditions

Many of the optimal gait patterns observed for models of varying levels of weakness were spatiotemporally asymmetric. Since reducing these asymmetries is a common rehabilitation objective for people post-stroke [62], we conducted a second set of optimizations to determine how reducing the step length and step time asymmetry would impact metabolic cost. Therefore, we added two terms to the objective function ($J_2$; Eq 6) that allowed us to reduce step time asymmetry or step length asymmetry for the different models.

$$J_2 = J_1 + (w_1 * SLS_{goal}) + (w_2 * STS_{goal}) \tag{6}$$

The step length symmetry goal ($SLS_{goal}$) was formulated to minimize the step length asymmetry (SLA; Eq 7) during the stride. Theoretically, we would want to compute the step length for the non-paretic ($SL_{NP}$) and paretic ($SL_P$) sides across the gait cycle for each iteration. However, this approach does not work well for predictive simulations because gradient-based optimization approaches require smooth, differentiable objective function terms in order to achieve reliable convergence. Therefore, we instead approximated step length asymmetry by performing the following steps. We first set the total stride length to be equal to the stride length of the respective nominal condition (i.e., the stride length from the 0.75 m/s speed with 20% weakness), such that the stride length is matched within a speed/model condition. Then, to target a symmetrical step length, we computed the length for the right and left step that would both result in symmetrical step lengths during the gait cycle. To enforce these step lengths during the optimization, we set penalties that accumulate if distance between the feet exceed the targeted step length distance for each foot. Therefore, we are indirectly setting the step length for the paretic and non-paretic step using smoothing functions that work well for the optimization algorithm (see S1 Text).

$$SLA = \frac{SL_{NP} - SL_P}{SL_{NP} + SL_P} \times 100 \tag{7}$$

The step time symmetry goal ($STS_{goal}$) was designed to reduce the step time asymmetry during the gait cycle (Eq 8). As with the $SLS_{goal}$, we needed to find an approximate form of step time asymmetry for this goal to work well within our optimization framework. Therefore, for this goal, we computed an approximation of step time asymmetry by detecting the number of time points each foot was in contact with the ground and computed the asymmetry index by normalizing by the total number of time points. Time points when both feet were in contact were assigned to the leading foot. Once we have the total number of nodes for non-paretic ($ST_{NP}$) and paretic step time ($ST_P$), respectively, we can then compute the asymmetry index based on the number of nodes and the stride time (see S1 Text).

$$STA = \frac{ST_{NP} - ST_P}{ST_{NP} + ST_P} \times 100 \tag{8}$$

Since each of these symmetry goals are approximations of step length and step time asymmetries, we then computed the actual asymmetry indices to check if the optimal solution achieves the intended symmetrical pattern. If there were large discrepancies, we would then be able to adjust settings on the SLS or STS goal and re-run the optimization. We chose to run this symmetry sub-analysis on the 0.75 m/s speed because the nominal results had a consistent pattern of increasing step length asymmetry with greater hemiparesis. We also chose this

speed for this analysis because it aligns closely with the average gait speed reported in recent studies for patients post-stroke [35]. First, we solved a new set of optimal control problems to reduce the step time symmetry for each model at 0.75 m/s. For the first set of optimizations, $w_1$ was set to 8 and $w_2$ was set to 0, such that reducing the step length asymmetry was not part of the objective, only step time asymmetry.

Finally, we solved another set of optimal control problems at 0.75 m/s with the added goal of reducing both step length and step time asymmetry. In this case, the weighting for $J_2$ were set as $w_1$ set to 8 and $w_2$ set to 5. These weightings were chosen heuristically such that we were able to reduce the asymmetry in each domain while getting good convergence. We noticed that if either $w_1$ or $w_2$ were too large, the optimization would not converge in a reasonable amount of time.

## 2.4 Optimization settings

All optimal control problems (26 <u>total conditions:</u> four models x five speeds, plus three more for step time symmetry and three more for step time and step length symmetry) were solved with 201 evenly spaced grid points, with Hermite-Sampson discretization [63], using CasADi [64] and IPOPT [65]. Termination settings for the optimization were: 1e-4 constraint violation tolerance and 1e-4 convergence tolerance. Optimizations were solved on a four-core desktop computer with a 3.3 GHz Intel Core i5-6600 processor. In total, we ran 60 optimizations for the nominal conditions (20 conditions with 3 initial guesses each), plus 18 additional optimizations for the symmetrical goal conditions. Each of the individual optimizations was solved on our desktop computer in times ranging from 2 to 14 hours, depending on the condition and initial guess.

## 2.5. Outcome measures and evaluations

The step length for the non-paretic (right) limb was calculated by taking the anterior-posterior (AP) distance between the ground-contact elements on the heels at the instant of non-paretic foot-strike where foot-strike was defined as the point when the vertical GRF was greater than 20 N [66]. The step length for the paretic (left) limb was calculated similarly at the instant of paretic foot-strike. Step time for the non-paretic limb was calculated as the time from paretic foot-strike to non-paretic foot strike, and step time for the non-paretic limb was calculated as the time from non-paretic foot-strike to paretic foot-strike.

The metabolic cost of transport (COT) for each condition was also estimated based on the *Umberger2010MuscleMetabolicsProbe* accessed through OpenSim [67,68]. The COT was calculated for each condition while holding muscle mass constant, to replicate weakness in the limb without muscle atrophy. In addition to predicting the change in COT between conditions, the positive and negative muscle fiber work performed for each muscle was also calculated (Eq 9), since muscle fiber work is a component of the metabolic cost model. First, the muscle fiber power was calculated for each muscle across time, then the positive and negative portions of the power curve were separately integrated to calculate positive and negative mechanical work, respectively for each limb. For the non-paretic side, positive mechanical power ($W_{NP}^+$) was calculated by:

$$W_{NP}^+ = \sum_{m=1}^{12} \int_0^{t_f} P_{NP_m}^+ dt \qquad (9)$$

where $m$ is each of the 12 muscles of the limb, $t_f$ equals the total time of the stride and $P_{NP_m}^+$ is the positive mechanical power for the $m$th muscle on the non-paretic side. Negative mechanical work was calculated based on the negative power of each non-paretic muscle. The positive and negative mechanical work for the paretic limb was calculated the same way as above, but

with power data from the paretic limb. As with the metabolic COT calculations, we normalized both the positive and negative mechanical work by dividing by displacement to compute the positive and negative mechanical COT.

Lastly as different metabolic cost models can result in different predictions of metabolic cost [69], we evaluated whether our results would change with a different metabolic cost model. Therefore, we computed the metabolic COT using the Bhargava model [70] for each of the nominal conditions to compare with the results from the Umberger model.

## 3. Results

### 3.1. Overview and model validation

Overall, the kinematic and GRF results for the base model across speeds shared broad similarities in patterns with previous experimental data, which gave us confidence that the modeling and optimization methods we used produced sensible results (Figs 2 and 3). For example, peak knee flexion angle occurred earlier in the gait cycle as speed increased (Fig 2B), which aligns with experimental results [71]. The peak ankle plantarflexion angle at push-off also occurred earlier in the gait cycle at faster speeds compared with slower speeds (Fig 2C), also matching with experimental data [71]. Generally, the GRFs also matched experimental data, with greater peak vertical GRFs (Fig 2D) and greater posterior GRFs during early stance (Fig 2E) for faster speeds [72]. Across all speeds, the step time and step length variables were each close to symmetrical for the base model, demonstrating that the optimal gait patterns for symmetrical models were symmetrical (Fig 4A and 4D).

### 3.2. Effects of simulated hemiparesis on spatiotemporal asymmetry

Generally, for models with muscle strength asymmetries, the optimal solution resulted in asymmetrical spatiotemporal patterns where the magnitude and direction of the asymmetries depended on both the level of muscle weakness and the gait speed (Fig 4A and 4B). For slower gait speeds (e.g., 0.25 and 0.50 m/s), the optimization resulted in a positive step time asymmetry for both the 40% and 60% weakness models which corresponds to the model taking longer to transition from the paretic to non-paretic limb than vice versa. At faster speeds (e.g., 1.00 or 1.25 m/s) the optimal gait pattern had a negative step time asymmetry for those same models. This suggests that the direction of step time asymmetry (positive or negative) is impacted by the gait speed that an individual walks and the degree of muscle weakness. Generally, the step times for the non-paretic side (Fig 4B) were about 0.40 seconds across most speeds and conditions, while the step times for the paretic side (Fig 4C) across the conditions were greater at faster speeds than slower speeds (i.e., ~0.40 seconds at slower speeds and ~0.50 seconds at faster speeds).

The direction and magnitude of step length asymmetry also varied depending on both the level of muscle weakness and the gait speed (Fig 4D). Marked step length asymmetries were generally observed once strength was reduced by 40%. For most speeds, the optimal solution for the model with 40% weakness was a positive step length asymmetry, with the largest asymmetry being observed in the weakest model at the slowest speed. These positive step length asymmetries correspond to a gait pattern where longer steps are taken with the paretic limb than the non-paretic limb. An exception to this trend occurred in the 1.25 m/s condition which resulted in a negative step length asymmetry. The optimal solutions produced greater step lengths on both the non-paretic (Fig 4E) and paretic (Fig 4F) sides for faster speeds compared with slower speeds. The 40% weakness model at 1.25 m/s resulted in step times and step lengths that stood out from other similar conditions, with a much shorter right step time and

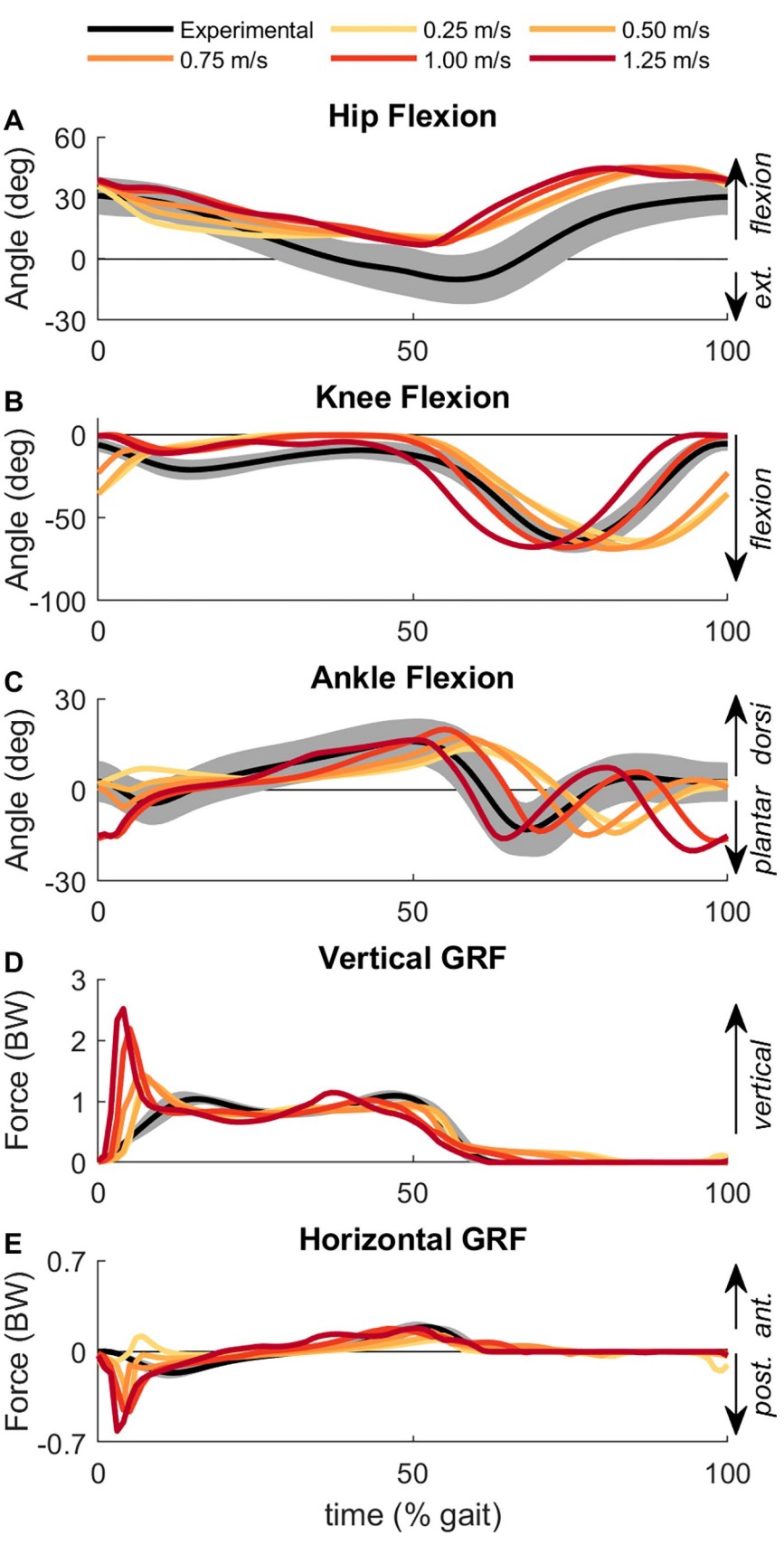

**Fig 2. Baseline kinematic and ground reaction forces.** The (A) hip flexion, (B) knee flexion, (C) ankle flexion angles and (D) vertical GRF and (E) horizontal GRF for the base model across each of the five gait speeds. Experimental kinematics and GRFs from adults walking at 1.1 m/s is shown as the mean (black line) and 2 standard deviations (shaded region) [85]. Time is normalized to the gait cycle with 0% being the time of right heel strike.

shorter right and left step lengths than other similar conditions (i.e., 40% weakness at 1.00 m/s or 60% weakness at 1.25 m/s).

### 3.3. Metabolic and mechanical cost of transport for nominal conditions

Despite the optimal solutions having marked spatiotemporal asymmetries, the computed metabolic COT was relatively consistent within speeds across different model weaknesses (Fig 5A). At speeds between 0.25 and 0.75 m/s, the metabolic COT was similar across all models, but there were subtle differences across the models in metabolic COT at 1.00 and 1.25 m/s. The metabolic COT was greater at 1.25 m/s than 1.00 in the 40% (3.57 J/kg/m at 1.00 vs 4.66 J/kg/m at 1.25 m/s) and 60% weakness (2.93 J/kg/m at 1.00 vs 3.55 J/kg/m at 1.25 m/s) models, while the metabolic COT was greater at 1.00 m/s than 1.25 in the symmetrical (3.55 J/kg/m at 1.00 vs 3.22 J/kg/m at 1.25 m/s) and 20% weakness models (3.56 J/kg/m at 1.00 vs 3.26 J/kg/m at 1.25 m/s). Gait speed had a substantial effect on the computed metabolic COT across the speeds we tested, with greater COT at slower gait speeds than faster gait speeds. At faster gait speeds (e.g., 1.00 and 1.25 m/s), the metabolic COT was between 3–4 J/kg/m, while at slower speeds (e.g., 0.25 and 0.50) the metabolic COT was close to 10 J/kg/m, which matches closely with the trend of metabolic COT across speeds in neurotypical individuals [15,59]. The consistency of the COT for a given speed resulted from complementary changes in limb-specific metabolic COT. Metabolic COT on the non-paretic limb increased with greater weakness levels (Fig 5B) while COT was reduced proportionally on the paretic limb (Fig 5C).

One important factor to consider when determining metabolic energy consumed by muscles is the amount of mechanical work done by each muscle during walking [73,74]. Therefore, we evaluated how positive and negative mechanical work changed across speeds and conditions. Overall, the optimal solutions resulted in gait patterns that had greater positive mechanical COT at slower gait speeds (Fig 5D). Additionally, for models with greater muscle weakness there was a decrease in the magnitude of the positive mechanical COT. While the non-paretic limb had a similar level of positive mechanical COT across the different models within a speed (Fig 5F), the reduction in total positive mechanical work was driven by a considerable decrease

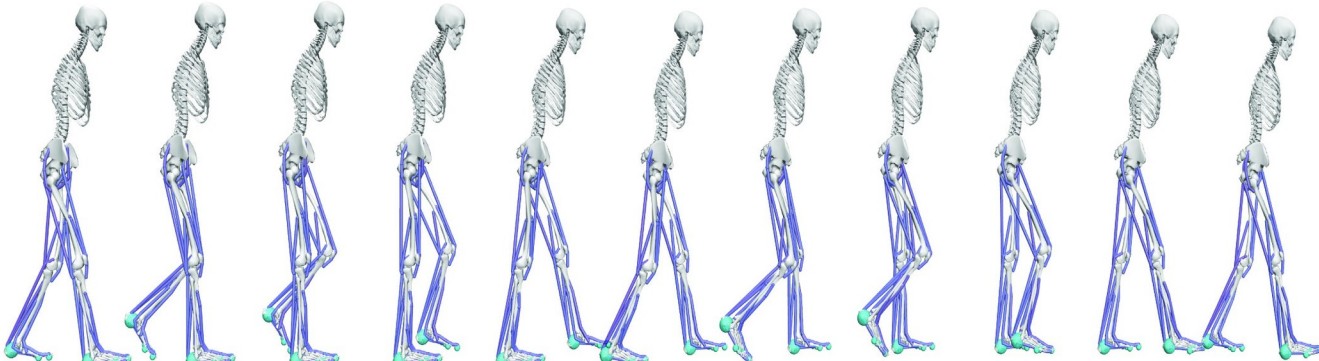

**Fig 3. Time lapse images.** Optimal result of model [86] walking at 1.00 m/s in the base condition. Blue line segments show muscle paths for the 18 muscles in the model, and the light blue spheres at the feet show the arrangement of the foot-contact elements. The OpenSim image depicted is based on models available at simtk.org and the specific model and data shown here can be downloaded from https://simtk.org/projects/post-stroke-sym.

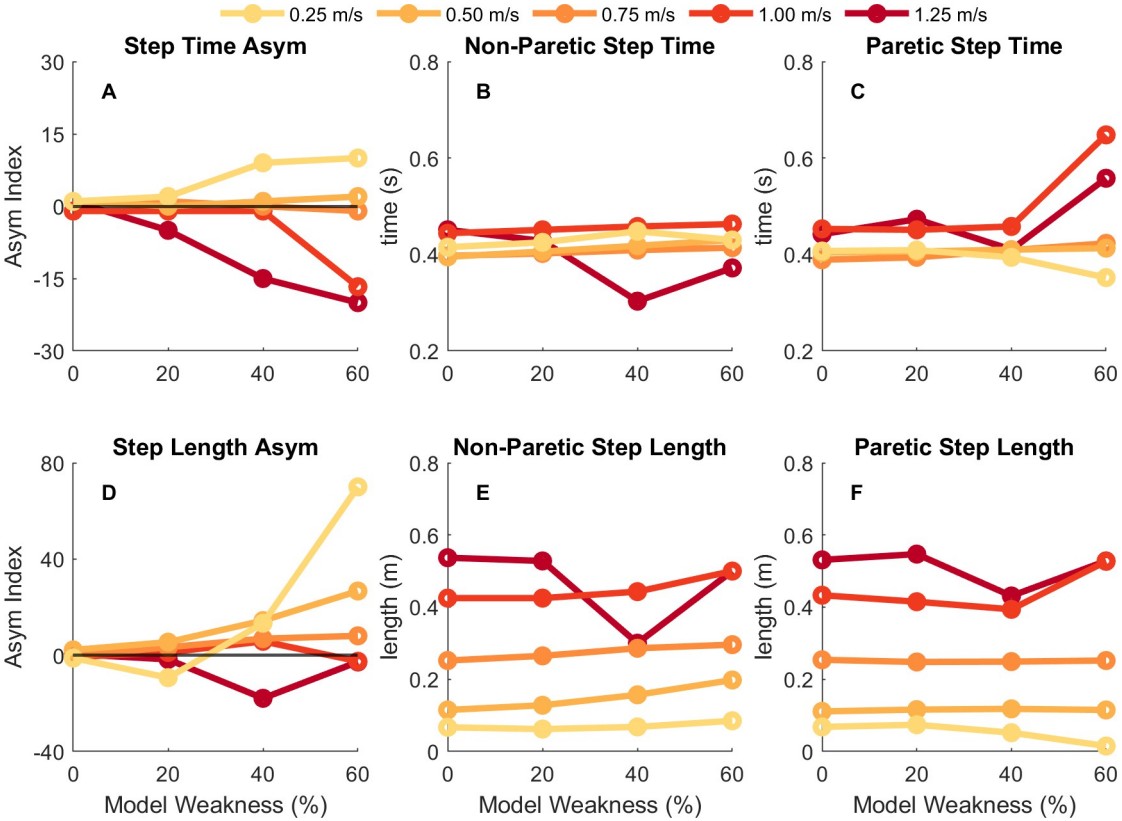

**Fig 4. Step time and step length results.** Spatiotemporal results for step time (top row) and step length (bottom row) for each condition and each speed. The leftmost column is the step time or step length asymmetry index (A, D), middle column is the non-paretic (right) step time or step length value (B, E), and rightmost column is the paretic (left) step time or step length value (C, F). Darker, red colors depict results for faster speeds while lighter, yellow colors depict results for slower speeds. Positive asymmetry indices indicate that the non-paretic value is greater than the paretic value, whereas negative asymmetry indices indicate that the paretic value is greater than the non-paretic value.

in the magnitude of positive mechanical COT by the paretic limb with increasing weakness (Fig 5F). These trends were also seen for negative mechanical COT, with greater negative mechanical COT at slower speeds and a decrease in negative mechanical COT with greater muscle weakness.

The second important factor in determining metabolic energy consumed by muscles is the level of muscle activation throughout the gait cycle. Total muscle activation increases at both faster gait speeds and greater levels of simulated hemiparesis (Fig 5G). At the individual limb level, the muscle activations for the non-paretic and paretic limbs increase with speed and weakness as well (Fig 5H and 5I). Overall, the metabolic energy cost does not significantly vary across level of simulated hemiparesis, which is likely a result of a decrease in mechanical muscle fiber work which offsets the increase in muscle activation with greater muscle weakness.

We observed a typical U-shaped pattern of metabolic COT with respect to gait speed within a model (Fig 6). Using these results, we can identify the "effort-optimal" solutions that resulted in the minimum metabolic COT within a model across the speeds. For both the base and 20% weakness models, the minimum metabolic COT occurred at a gait speed of 1.25 m/s, which is similar to what is observed experimentally in neurotypical individuals [59,75,76]. For the 40% and 60% weakness models, the minimum metabolic COT instead occurred at a slower speed of 1.00 m/s.

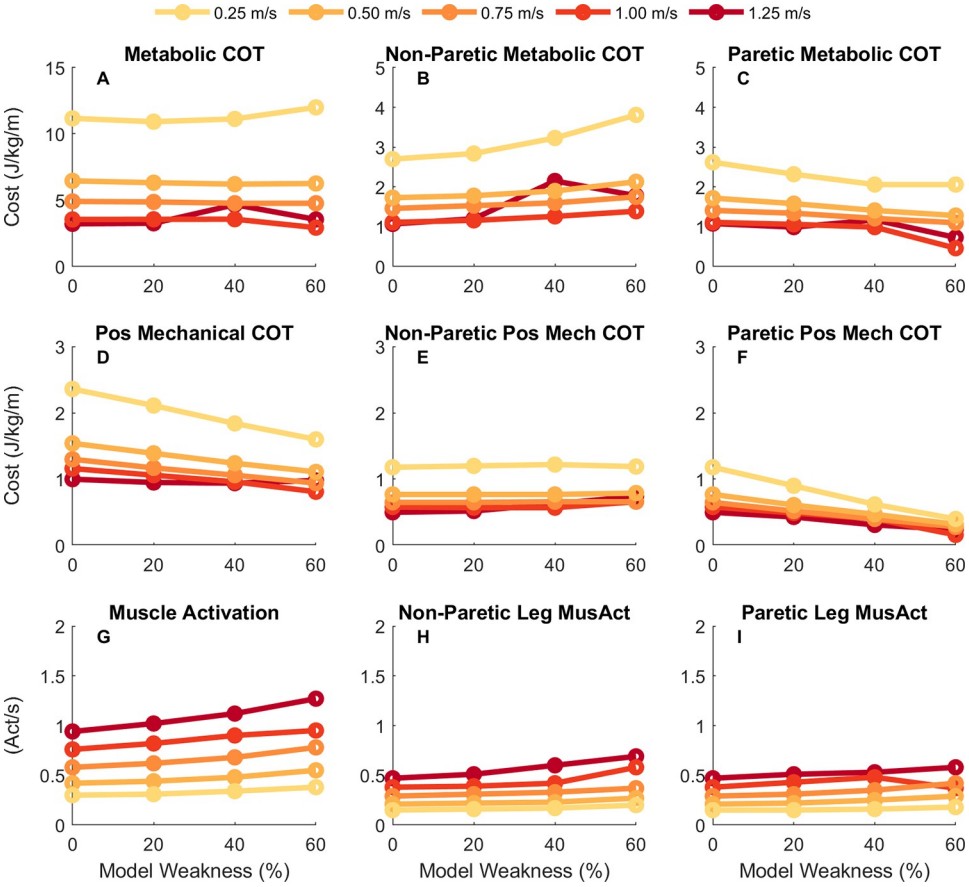

**Fig 5. Metabolic and mechanical cost of transport.** Metabolic cost of transport (COT; top row), positive mechanical COT (middle row), and sum of the integrated muscle activation (bottom row) for each condition. The leftmost column shows the result for the sum across both limbs, while the middle column shows the results for the non-paretic (right) limb and the rightmost column shows the result for the paretic (left) limb. Darker, red colors are for faster speeds while lighter, yellow colors are for slower speeds. Metabolic COT remained relatively consistent within a speed across the different weakness models, partially a result of an increase in mechanical work done by the right (non-paretic) limb with a proportional decrease in mechanical work done by the left (paretic) limb.

To evaluate the sensitivity of our results and conclusions to the metabolic model chosen, we computed the metabolic COT using the Bhargava metabolic model [70]. The Bhargava model resulted in slightly greater metabolic COT (average offset of $0.37 \pm 0.17$ J/kg/m) than the Umberger model, but the trends across the conditions were similar which suggests that our conclusions would not change with either the Umberger or Bhargava energetics models.

## 3.4. Effects of minimizing step length and step time asymmetry in a model of hemiparesis

Finally, we evaluated whether the predicted metabolic COT would change when enforcing step length or step time symmetry for models with simulated hemiparesis. To perform this analysis, we added a term to the objective function which produced gait patterns with approximately symmetrical step lengths across the different hemiparetic models at a speed of 0.75 m/s. The optimal solutions for the 20%, 40%, and 60% models had step length asymmetry indices less than 1.5%, which was close to the symmetrical target of 0% asymmetry (Fig 7B). However, this resulted in greater step time asymmetry for these gait patterns (Fig 7A). Minimizing step

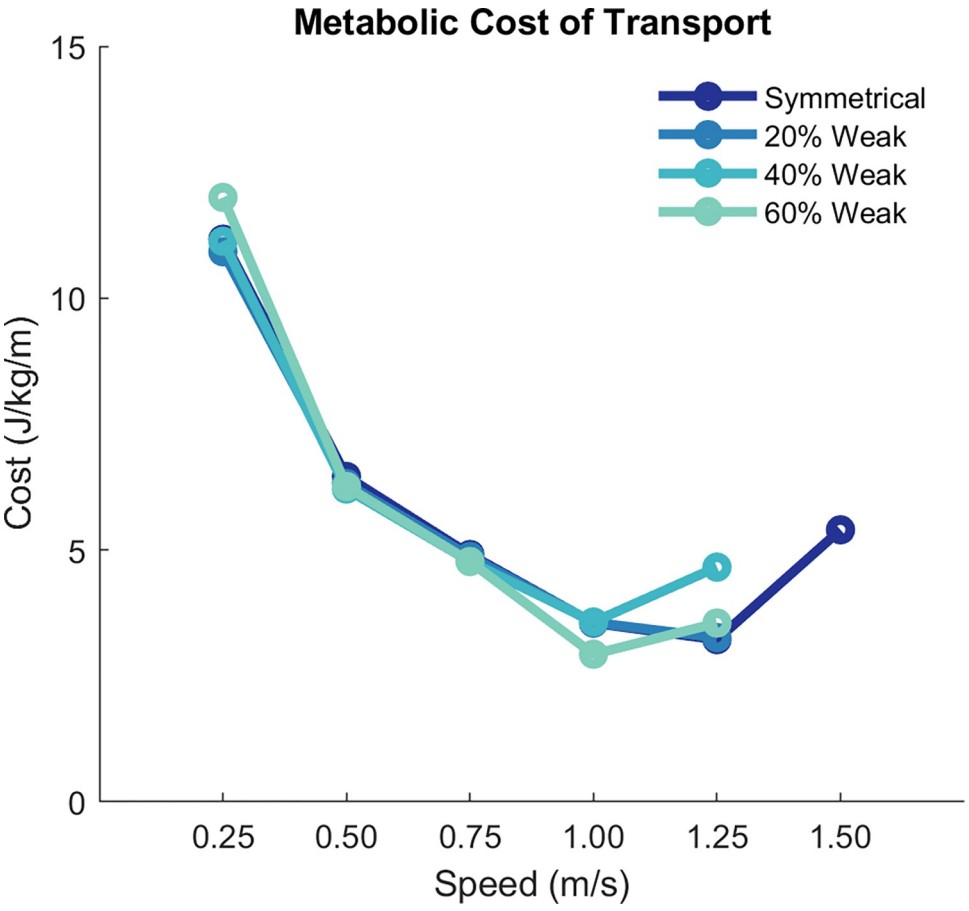

**Fig 6. Metabolic cost of transport across speeds.** Dark blue represents results for the symmetrical (or base) model, while blue-green lighter colors represent the asymmetrical models. The symmetrical condition included an optimization at 1.50 m/s so that we could establish the U-shaped curve of metabolic cost (where 1.25 is approximately the metabolic-optimal speed for the symmetrical model). We did not solve the optimal control problem for other models at 1.50 m/s because the weakness in the 40% and 60% models prevented convergence at speeds faster that 1.25 m/s.

length asymmetry resulted in a slightly greater metabolic COT than the nominal conditions by 1–4% (Fig 7C). We then added a third term to the objective function, which in combination with the others, resulted in gait patterns that had approximately symmetrical step lengths (maximum asymmetry = 2.6%; Fig 7B) and step times (maximum asymmetry = -1%; Fig 7A). Despite adopting a gait that was nearly symmetric overall, the metabolic COT for these conditions was less than 1% greater compared to when step length asymmetry was minimized alone, and only deviated by 5% from the nominal conditions.

## 4. Discussion

### 4.1. Overview of key results

The purpose of this study was to quantify the spatiotemporal asymmetries and changes to metabolic cost that emerge from effort-optimal predictions of gait with models of simulated hemiparesis across a wide range of speeds. Predicting the optimal gait pattern for models with simulated hemiparesis allows us to gain insight into the isolated effect of unilateral muscle weakness on gait asymmetry and metabolic COT. We found that the magnitude and direction

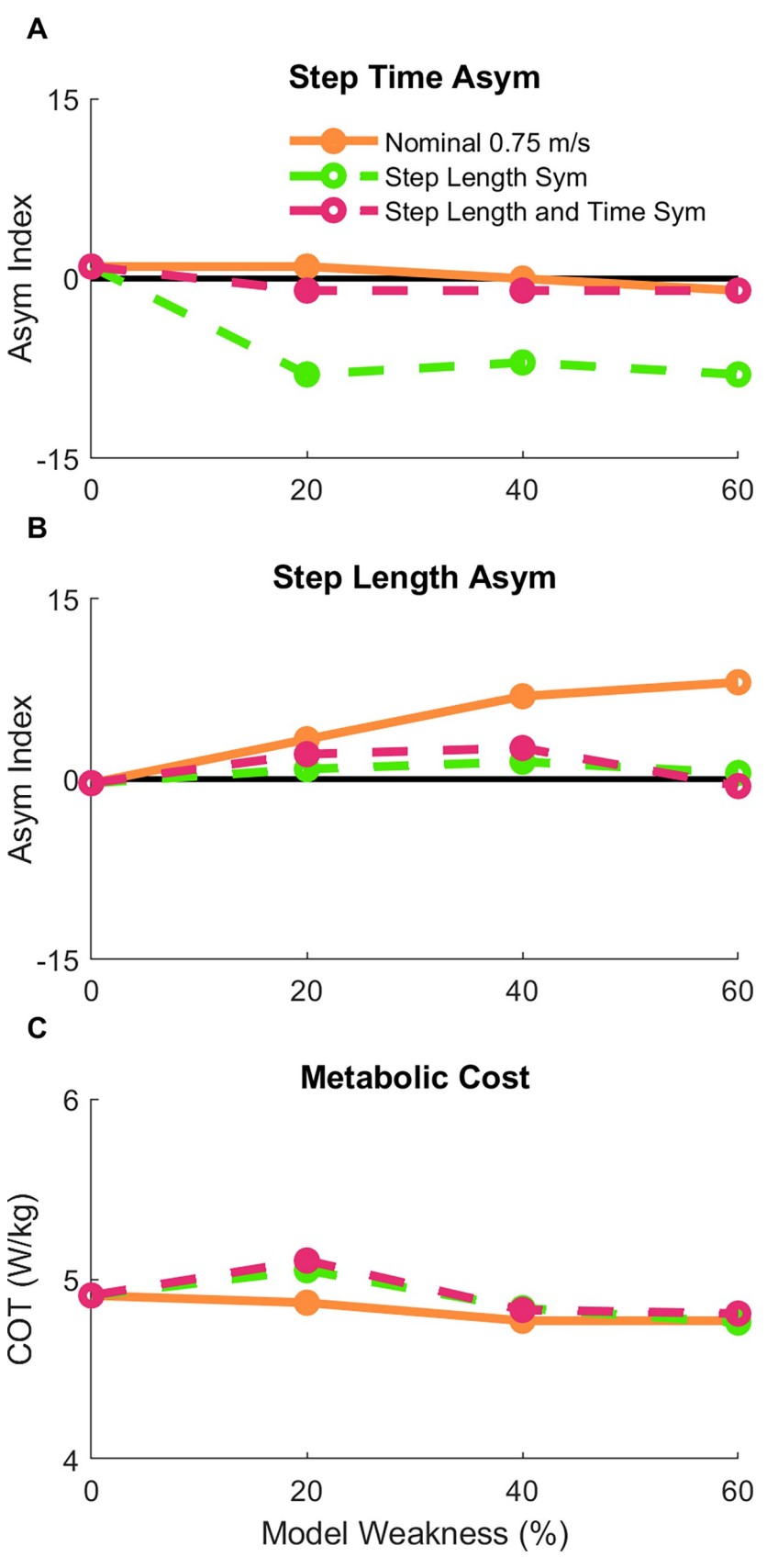

**Fig 7. Secondary analysis of spatiotemporal asymmetry.** After the nominal conditions (orange solid line), additional optimizations were performed across the three hemiparetic models with the goal of reducing the step length asymmetry (green dashed line) to assess the effect of enforced step length (B) symmetry on step time asymmetry (A) and metabolic cost (C). Finally, a set of optimizations were performed with the goal of reducing both step length and step time asymmetry (purple dashed line).

of spatiotemporal asymmetry depended on the level of muscle weakness as well as the gait speed, with greater asymmetries corresponding to a greater level of muscle weakness. We then compared the metabolic COT across all optimal gait patterns to assess whether simulated hemiparesis and spatiotemporal asymmetries correspond with an increase in metabolic cost. We found that the predicted metabolic COT varied little with increasing levels of weakness, but changed with speed as expected based on experimental data, e.g. [47]. Finally, we evaluated whether enforcing step length or step time symmetry for the hemiparetic models would increase the metabolic COT. The metabolic COT only changed by a maximum difference of 5% between the nominal and symmetrical conditions. Overall, our results suggest that the spatiotemporal asymmetries seen in people post-stroke might be derived in part from optimal adaptations to underlying impairments such as hemiparesis. However, the presence of gait asymmetries does not necessarily contribute to the increase in metabolic cost that is observed relative to neurotypical individuals.

Overall, a greater level of hemiparesis resulted in a more asymmetrical gait pattern in both step time and step length across all speeds, however the direction of the asymmetry depended on the gait speed. The 20% weakness model had step time asymmetries ranging from -5 to 2% and step length asymmetries ranging from -10 to 6%, while the 60% weakness models had step time asymmetries ranging from -20 to 10% and step length asymmetries ranging from -3 to 70%. This suggests that the effort-optimal gait patterns for individuals with mild unilateral strength deficits may be closer to symmetrical, compared to individuals with much greater muscle strength impairments who may have effort-optimal gait patterns with greater spatiotemporal asymmetries.

We also found that metabolic COT is relatively consistent across levels of simulated hemiparesis. Metabolic cost is impacted by the level of muscle activation and the magnitude of mechanical work done throughout the movement. While muscle activation increased with greater levels of simulated hemiparesis, the positive and negative work performed by the muscle fibers decrease with greater levels of simulated hemiparesis. The decrease in work performed by the muscle fibers is driven by a decrease in work done by the paretic limb due to less muscle force generation. Overall, this effectively creates a trade-off between muscle activation and mechanical work, resulting in a similar level of metabolic cost across models within a gait speed.

There were substantial speed effects on both spatiotemporal asymmetry and metabolic cost. The magnitude and direction of step length asymmetry was also dependent upon the gait speed. For hemiparetic gait, a positive step length asymmetry indicates shorter steps with the paretic limb than the non-paretic limb. This pattern can be expected of a person post-stroke if they are unable to swing their paretic limb forward. Many of the effort-optimal gait patterns fall into this category of having positive step length asymmetry with a couple of exceptions at the 0.25 and 1.25 m/s conditions. For the 60% weak model, there were greater positive step length asymmetries at slower speeds than for faster speeds. The overall task demands for the slow speed may allow the model to take extremely short paretic side steps, while still matching the task speed. The heterogeneity and relative ranges of both positive and negative step length asymmetries largely mirrors what has been measured in previous experiments of people post-stroke [35,77,78].

Our results suggest that the direction of step time asymmetry depends on the gait speed rather than the level of hemiparesis. At 1.25 m/s speed, the effort-optimal gait patterns resulted in negative step time asymmetries across the three hemiparetic models, while for the 0.25 m/s speed, the effort optimal gait patterns result in positive step time asymmetries. Paretic step time here was defined as the time from non-paretic foot strike to paretic foot strike, so it contains the duration of the double support time with the leading, non-paretic limb and the paretic side swing time. Therefore, a negative step time asymmetry (greater paretic step time than non-paretic step time) would suggest that the model is using its non-paretic limb for body weight support and propulsion for a greater period than the paretic limb, which is what one might expect since weakness occurs in the paretic limb. In contrast, a positive step time asymmetry may be beneficial at the slow speeds because the paretic swing limb may be less able to take advantage of passive dynamics relative to what happens at faster speeds, resulting in a greater activation during swing phase to swing the leg forward.

## 4.2. Clinical implications for post-stroke rehabilitation

The results from our simulation study have generated a couple of testable hypotheses. First, that the direction of spatiotemporal asymmetries depends on the severity of hemiparesis and walking speed. We know that spatiotemporal asymmetries, such as step length asymmetry, in people post-stroke are heterogeneous, with some individuals having longer step lengths with their paretic limb than their non-paretic and others having shorter step lengths with their paretic limb than non-paretic limb [35,78], however what drives these responses remains unclear [77]. Our results suggest that one of the driving factors of asymmetry could be the level of hemiparesis, which could be tested via an experimental study by measuring the relationship between muscle strength asymmetries and spatiotemporal asymmetries during gait at different speeds.

Furthermore, the hypothesized relationship between severity of hemiparesis and the effort-optimal gait patterns (i.e., direction and magnitude of spatiotemporal asymmetries) suggests that clinical rehabilitation programs could be individually tailored based on measures of between-limb differences in strength. Typical goals for gait rehabilitation programs for people post-stroke are to have the individuals walk more symmetrically [62,79]. However, it's still an open question whether this goal could be considered the optimal way to walk for people post-stroke, with recent data suggesting that symmetrical step lengths do not improve measures of metabolic COT [33,35] and result in other kinematic and kinetic asymmetries [32]. The results of our predictive simulations are consistent with these studies: when we enforced step length symmetry for the 0.75 m/s speed, the resulting gait pattern had significant step time asymmetry, with a small increase in metabolic COT. The experimental results in combination with our simulated results are reasonable given the underlying anatomical system is asymmetric, therefore optimal gait patterns are also likely to be asymmetric. However, that the symmetry conditions resulted in only small increases in metabolic COT suggest that there could be multiple gait strategies for people post-stroke that result in similar consumption of metabolic energy. Therefore, patients who prioritize a symmetrical appearance of gait may be able to perform their preferred gait pattern without a large penalty on their endurance.

Another important research question for gait in people post-stroke is what are the factors that contribute most to the slower gait speed observed in this population compared to age-matched controls [26,35,80,81]? Our data suggest that moderate levels of hemiparesis could contribute to the reduction in the energy optimal gait speed, as minimum metabolic COT occurred at 1.00 m/s for the 40% and 60% weakness models compared with a minimum at 1.25 m/s for the base and 20% weak models. However, this slight reduction in speed does not

match the difference in preferred gait speed in people post-stroke relative to control (~0.7 m/s [35]). So, while unilateral muscle weakness may partially contribute to the reduction in gait speed, other types of impairments such as spasticity and abnormal muscle co-activation patterns or different priorities during gait (e.g., balance and comfort) may have additional contributions.

Lastly, our data present another potential explanation (besides metabolic COT) for slower walking speed in people post-stroke: preferred gait speed for people post-stroke may also be impacted by an individual's desire to minimize observable spatiotemporal asymmetries. Our data suggest that moderate speeds of 0.50 and 0.75 m/s result in effort-optimal gait patterns with moderate levels of gait asymmetry (<10% asymmetry indices), while slower (0.25 m/s) or faster speeds (1.00 or 1.25 m/s) can result in levels of gait asymmetry greater than 10%. Therefore, we can hypothesize that people post-stroke may walk with slower gait speeds since this allows them to maintain lower gait asymmetry while also keeping metabolic COT low. Further explanation of the preferred gait patterns in people post-stroke could be explained by factors like balance or comfort, though how people post-stroke sense and perceive these factors is an important question to be addressed but is beyond the scope of this study.

### 4.3. Modeling decisions and limitations

Our results build upon several modeling choices that we made throughout the study, and as such, there are a few limitations to our project that should be considered in future research. First, we constrained our model to move only in the sagittal plane because our primary kinematic variables of interest for this project were step length and step time asymmetry. Since these asymmetries result from deviations in the sagittal plane, restricting the model to move in the sagittal plane only allowed us to reduce the number of confounding variables in our analysis. In the future, expanding upon this work with a fully three-dimensional model would allow for greater insight into non-sagittal plane compensations that can occur in people post-stroke, such as hip hiking or hip circumduction.

Additionally, we chose an objective function that minimizes the integrated sum of muscle excitations cubed across all muscles, which has been proposed to be a representation of minimizing muscle fatigue, or maximizing muscle endurance [40]. This objective is formulated from experimental data that suggests the muscle force-endurance relationship is approximately cubic [55,82]. While we evaluated the predicted metabolic COT from the results, we did not use metabolic COT in the objective function. Therefore, there could be different gait patterns that would further reduce the metabolic cost across each of the conditions. Additionally, co-activation of muscles and muscle spasticity, which often presents together with hemiparesis, can also affect the gait performance in individuals with neuromuscular impairments. Co-activation of muscles could be considered a constraint that emerges due to reliance on descending control pathways that couple muscles within and across joints, e.g. [13,14]. However, it is conceivable that reducing effort remains an important objective in post-stroke gait even in the presence of this obligatory co-activation. While people with hemiparesis do exhibit other impairments, this study focused on a single impairment (unilateral weakness) to develop the specific correlations between the variables: weakness, asymmetry, and metabolic energy. While different objective functions will result in different gait patterns from an optimal control solution [40,48,83], we believe minimizing muscle excitations cubed is appropriate for this study as fatigue is likely to be especially relevant for people with neuromuscular impairments.

Overall, the kinematics and ground reaction forces for our results match well with experimental gait patterns for neurotypical adults (Fig 2), however there were a couple deviations from experimental data observed in our results. During mid-stance, the model has a nearly

fully-extended knee (~0˚; Fig 1B), while typical human gait patterns involve some mild knee flexion during mid-stance (~10–20˚; [71]). It is possible that using an objective function that weights muscle excitations times muscle volume (rather than only muscle excitations) could induce more knee flexion during stance (e.g., [40,84]). The other kinematic deviation that occurs is at the hip angle, where the model never achieves hip hyperextension during gait, as is typically seen in human subjects. This is likely due to a modeling choice to have the pelvis and trunk locked together (e.g., no lumber flexion angle) to reduce the total number of degrees-of-freedom in the model and increase the computational speed for each optimization. The consequence of this choice was that both the lumbar and pelvis were tilted forward throughout the stride (Fig 3), which resulted in an increase in the hip flexion angle relative to the femur.

Finally, the modeling of hemiparesis in our project involved several decisions. First, we decided to model hemiparesis instead of other impairments such as abnormal muscle coordination patterns or other changes in muscle properties because hemiparesis is straightforward to model by reducing peak muscle forces in the model. Future work should add other impairments, either on their own or in combination with hemiparesis, to better understand the independent or combinatory effects of the array of impairments after a stroke. Further, we decided to simulate hemiparesis by reducing the peak isometric muscle forces instead of modifying the maximum excitation values, which would relate to the decrease in central drive in people post-stroke. Modeling decreased central drive in a predictive simulation paradigm is impractical from a computational standpoint, as it would only affect the results when the maximum allowable excitation in the simulated weakness condition is exceeded in the base conditions. In gait, most muscles operate in a submaximal state, oftentimes far below full excitation. Therefore, setting a threshold excitation of 60% of baseline, for example, would likely result in no changes to the gait strategy since none of the muscles surpassed 60% excitation level during the baseline stride. Furthermore, modeling decreased central drive is challenging from a physiological standpoint because it is difficult to assess the magnitude of impairment in central drive *in vivo*, instead, it's much simpler to measure muscle strength in the limbs using a dynamometer (e.g., [10]). Therefore, our choice of reducing peak isometric muscle force as a representation of a reduction in the ability to produce force allowed for a reasonable way to model hemiparesis and is applicable since measuring the magnitude of impairment can be easily done in clinical settings. While we modeled a uniform level of muscle weakness within each hemiparetic model (i.e., all muscles on the left limb were reduced by the same magnitude) future work could study how different combinations of muscle weakness affect gait performance. Another important modeling decision was to keep muscle mass constant across all hemiparetic models to simulate muscle weakness without muscle atrophy. Muscle mass is used for computing metabolic COT, because the metabolic energy consumed by muscles depends on activation levels and the volume of muscle activated [68]. If we instead modeled muscle weakness alongside muscle atrophy, it would result in a reduction in metabolic cost for the hemiparetic models compared with our results due to a reduction in activated muscle mass.

## 4.4. Conclusion

In this study, we predicted the effort-optimal spatiotemporal patterns for gait with simulated hemiparetic musculoskeletal models. We found that the magnitude and direction of spatiotemporal asymmetry is affected by the level of hemiparesis and the gait speed, which may explain the well-known heterogenous distribution of spatiotemporal asymmetries observed in clinical data. However, the greater metabolic COT observed in people post-stroke compared to controls does not appear to be driven by hemiparesis, and instead may be driven by factors like muscle co-activation or abnormal muscle synergy patterns. Further, our data predict that

hemiparesis is one aspect that could lead to slower self-selected gait speeds in people post stroke, but other neuromuscular impairments or preferences may drive gait speed even slower than what was predicted to be energy optimal in our simulations. Lastly, our data provide additional theoretical support for the idea that asymmetrical gait patterns may be optimal when aspects of the underlying control system is asymmetrical. Overall, our study is a step towards a better understanding of how specific impairments in people post-stroke affect gait patterns and metabolic COT. Since it is difficult to study how distributed, unilateral muscle weakness alone affects gait with human participants, our predictive modeling approach can allow for these tests since we can build custom models that represent the type of impairment we are focused on testing. Future studies should extend the work presented here to explore the effects of additional impairments and build a more comprehensive understanding of how a range of impairments influence post-stroke gait.

## Supporting information

**S1 Text. The supporting information S1 Text gives full details on the step length symmetry and step time symmetry goals that we have introduced in this manuscript.**
(DOCX)

## Acknowledgments

The authors thank Natalia Sánchez for discussions on the content of the paper.

## Author Contributions

**Conceptualization:** Russell T. Johnson, James M. Finley.

**Data curation:** Russell T. Johnson.

**Formal analysis:** Russell T. Johnson.

**Funding acquisition:** James M. Finley.

**Investigation:** Russell T. Johnson.

**Methodology:** Russell T. Johnson, Nicholas A. Bianco, James M. Finley.

**Project administration:** Russell T. Johnson.

**Resources:** James M. Finley.

**Software:** Russell T. Johnson, Nicholas A. Bianco.

**Supervision:** James M. Finley.

**Visualization:** Russell T. Johnson.

**Writing – original draft:** Russell T. Johnson.

**Writing – review & editing:** Russell T. Johnson, Nicholas A. Bianco, James M. Finley.

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
