## [Decision Letter · Decision Letter 0]

7 Mar 2022

Dear Dr. Johnson,

Thank you very much for submitting your manuscript "Simulated hemiparesis increases optimal spatiotemporal gait asymmetry but not metabolic cost" for consideration at PLOS Computational Biology.

As with all papers reviewed by the journal, your manuscript was reviewed by members of the editorial board and by several independent reviewers. In light of the reviews (below this email), we would like to invite the resubmission of a significantly-revised version that takes into account the reviewers' comments. Please take particular care in the use of "stroke" as description of the model used here, it may suggest a stronger statement than the paper can make in its present form.

We cannot make any decision about publication until we have seen the revised manuscript and your response to the reviewers' comments. Your revised manuscript is also likely to be sent to reviewers for further evaluation.

Sincerely,

Aldo A Faisal

Associate Editor

PLOS Computational Biology

Mark Alber

Deputy Editor

PLOS Computational Biology

Reviewer's Responses to Questions

**Comments to the Authors:**

Reviewer #1: In this study, the investigators used 2D musculoskeletal modeling to investigate relationships between muscle strength, metabolic cost, and spatiotemporal gait symmetry. The results suggest that asymmetric walking patterns may be energetically optimal for persons with unilateral lower extremity weakness. This manuscript will be of interest to the biomechanics community and the readership of PCB. My comments and suggestions are included below:

ABSTRACT

In the abstract and throughout the manuscript, I recommend focusing on unilateral weakness rather than stroke. After reading the first sentence of the abstract, I was surprised that this manuscript used a model developed based on musculoskeletal parameters of healthy gait. While unilateral weakness is clearly relevant to stroke, I find the conclusions drawn about stroke gait to be a bit speculative given that many other common post-stroke deficits (e.g., spasticity, muscle atrophy, muscle stiffness, changes in tendon morphology, etc) are not addressed in the modeling and could significantly affect interpretations of this line of work. To this reviewer, it seems more appropriate to focus on the impact of unilateral weakness on interactions between cost and asymmetry while relegating potential applications to stroke to the discussion.

This comment is also relevant to the title, given that conditions other than paresis (e.g., fatigue) could a reduced ability to generate muscle forces unilaterally. I suggest revising to “Simulated unilateral weakness increases…”

INTRODUCTION

Related to my comment above, I recommend that the authors reframe the introduction around unilateral weakness rather than stroke.

It would be helpful if the authors could provide some context in the introduction about why weakness in particular – of the many common unilateral gait deficits – was chosen. I also think it is important to provide some background on prior studies of muscle weakness and their relationships to the parameters of interest here (e.g., asymmetry, energy cost). As currently written, the topic of weakness is not addressed at any length until the final paragraph of the introduction.

Expansion on prior findings where musculoskeletal modeling has led to insight into the effect of impairments on gait mechanics and energetics (line 72) would be helpful.

METHODS

Why was the DeGroote model chosen over the many other available musculoskeletal models?

The choice of a 2D model makes extrapolation of the findings to neurologic populations (where there are commonly out-of-plane compensatory movements) difficult. What was the justification for selecting a 2D model? This is justified somewhat in the limitations section of the discussion, but I advise caution in justifying such choices based on what was best for the modeling/software (e.g., time cost of simulations) vs. what was best for answering the research question. This also applies to the justification for the investigation of weakness in the discussion (“…because hemiparesis is straightforward to model”).

What was the rationale for systematically reducing all muscles by the same relative magnitude? Is there scientific justification for this in clinical work?

Given that co-contraction and spasticity are common in post-stroke gait (and other neurologic conditions with unilateral weakness), is it an appropriate optimization technique to minimize muscle excitations?

RESULTS

I am curious about the relationships between the simulated muscle activity and the spatiotemporal asymmetries, in particular the nonmonotonic relationships between step time asymmetry, weakness, and speed. It seems as though there must be conditions where weakness in certain muscle groups (e.g., plantarflexors) must drive the spatiotemporal and kinematic gait changes to a larger degree than others. Have the authors considered investigating any of these types of relationships?

The interactions between speed and cost are similarly interesting given their nonmonotonic trends. Is there any explanation for these relationships? It seems unusual and unexpected that the cost decreased with weakness when walking at 1.0 m/s (if I am interpreting lines 303-305 correctly).

Are there any GRF results to report? Given the relevance of the AP GRFs in neurologic populations, it would be interesting to see these data if available. These data would also be a nice complement to the mechanical work data.

DISCUSSION

The discussion is comprehensive and well-written. I do not have any revisions to suggest beyond those that may be needed in addressing the comments above.

Reviewer #2: Uploaded as attachment

**Have the authors made all data and (if applicable) computational code underlying the findings in their manuscript fully available?**

Reviewer #1: Yes

Reviewer #2: Yes

PLOS authors have the option to publish the peer review history of their article (what does this mean?). If published, this will include your full peer review and any attached files.

Reviewer #1: No

Reviewer #2: No
---

## [Decision Letter · Decision Letter 1]

3 Aug 2022

Dear Dr. Johnson,

We are pleased to inform you that your manuscript 'Patterns of asymmetry and energy cost generated from predictive simulations of hemiparetic gait' has been provisionally accepted for publication in PLOS Computational Biology.

Best regards,

Aldo A Faisal

Associate Editor

PLOS Computational Biology

Mark Alber

Deputy Editor

PLOS Computational Biology

Reviewer's Responses to Questions

**Comments to the Authors:**

Reviewer #1: The authors have done a thorough job in responding to my previous comments and suggestions. This manuscript will make a nice addition to the literature, and I thank the authors for sharing their interesting work.

Reviewer #2: All my comments have been addressed. Thank you.

**Have the authors made all data and (if applicable) computational code underlying the findings in their manuscript fully available?**

Reviewer #1: None

Reviewer #2: Yes

PLOS authors have the option to publish the peer review history of their article (what does this mean?). If published, this will include your full peer review and any attached files.

Reviewer #1: No

Reviewer #2: No

---

## [Editor Report · Acceptance letter]

2 Sep 2022

PCOMPBIOL-D-22-00011R1 

Patterns of asymmetry and energy cost generated from predictive simulations of hemiparetic gait

Dear Dr Johnson,

I am pleased to inform you that your manuscript has been formally accepted for publication in PLOS Computational Biology. Your manuscript is now with our production department and you will be notified of the publication date in due course.

With kind regards,

Anita Estes
